# Clinical manifestations and outcome of patients with primary amoebic meningoencephalitis in Pakistan. A single-center experience

**Shakeel Ur Rehman**[1], **Salman Farooq**[2]*, **Muhammad Bilal Tariq**[3], **Nosheen Nasir**[1], **Mohammad Wasay**[1], **Sobia Masood**[2], **Musa Karim**[2]

**1** Aga Khan University and Hospital (AKUH), Karachi, Pakistan, **2** National Institute of Cardiovascular Diseases (NICVD), Karachi, Pakistan, **3** University of Texas Health Science Center at Houston, Houston, TX, United States of America

* salmanfarooq8@gmail.com

**Data Availability Statement:** The data will be available on following contact: The Contact information of Institutional Research Ethics

## Abstract

Primary amoebic meningoencephalitis (PAM) is a rapidly progressing central nervous system (CNS) infection caused by *Naegleria fowleri*, a free-living amoeba found in warm freshwater. The disease progression is very rapid, and the outcome is nearly always fatal. We aim to describe the disease course in patients admitted with PAM in a tertiary care center in Karachi, Pakistan between the periods of 2010 to 2021. A total of 39 patients were included in the study, 33 males (84.6%). The median age of the patients was 34 years. The most frequent presenting complaint was fever, which was found in 37 patients (94.9%) followed by headache in 28 patients (71.8%), nausea and vomiting in 27 patients (69.2%), and seizures in 10 patients (25.6%). Overall, 39 patients underwent lumbar puncture, 27 patients (69.2%) had a positive motile trophozoites on CSF wet preparation microscopy, 18 patients (46.2%) had a positive culture, and 10 patients had a positive PCR. CSF analysis resembled bacterial meningitis with elevated white blood cell counts with predominantly neutrophils (median, 3000 [range, 1350–7500] cells/µL), low glucose levels median, 14 [range, 1–92] mg/dL), and elevated protein levels (median, 344 [range, 289–405] mg/dL). Imaging results were abnormal in approximately three-fourths of the patients which included cerebral edema (66.7%), hydrocephalus (25.6%), and cerebral infarctions (12.8%). Only one patient survived. PAM is a fatal illness with limited treatment success. Early diagnosis and prompt initiation of treatment can improve the survival of the patients and reduce mortality.

## Introduction

*Naegleria fowleri* is the only pathogenic protozoa in the Genus Naegleria. This amoeba-flagellate was discovered in 1899, but it was not until 1970s that the link between *Naegleria Fowleri* and primary amoebic meningo-encephalitis (PAM) was identified [1, 2]. *Naegleria fowleri* has 3 stages in its life cycle: cyst, trophozoite and flagellate, with trophozoite being the only infective stage. The protozoa penetrate the nasal tissue and migrating to the brain via the olfactory

Committee are: Tel: +92 21 3493 0051 EXT:4988/
2445 Email: erc.pakistan@aku.edu.

**Funding:** The authors received no specific funding
for this work.

**Competing interests:** The authors have declared
that no competing interests exist.

nerves causing primary amoebic meningo-encephalitis [1, 3]. Rarely, it can also spread in patient undergoing solid organ transplantation due to the immunosuppressed state of these patients. PAM generally affects children and young adults. In a case series of 142 patients done in the United States between 1937 and 2018 [4], the median age of the patients was 12 years (83% aged ≤18 years); males (76%) were predominately affected.

The acquisition of this infection is linked to very specific risk factors such as engaging in recreational activities involving swimming in freshwater lakes, river, canals, swimming pools, and during ablution as a part of religious practices [1, 5, 6]. Drinking infected water does not lead to the development of PAM.

After an incubation period of 7 days, patients experience fulminant, necrotizing, and hemorrhagic meningo-encephalitis. Initial symptoms mimic that of bacterial meningitis, characterized by severe headaches, stiff neck, fever (38.5˚C–41˚C), altered mental status, seizures, and coma, leading almost always to death with a fatality rate of over 95% [7, 8].

This study aims at identifying the spectrum of symptoms with which patients present to the hospital, reviewing the clinical course, duration of symptoms and lab parameters, specifically looking at the factors that predict poor outcome and the factors that delay the disease process and ultimately mortality.

## Materials and methods

An observational study was conducted on patients hospitalized with clinical features consistent with PAM at Aga Khan University Hospital (AKUH) between the periods of January 2010 to December 2021. In this case series, patients were included if the diagnostic workup was suggestive of PAM and who had received empiric treatment for PAM. Data was collected on clinical signs and symptoms, diagnostic workup including laboratory parameters and brain imaging findings, clinical course, and treatment outcome.

All patients who underwent lumbar puncture were tested for motile trophozoites on CSF wet preparation microscopy, CSF culture and PCR. Brain imaging findings were reported by expert neuroradiologist, and interval imaging comparisons were also made in patients where repeat brain scans were done. The study received approval from the Ethical Review Committee under # 2021-6074-18800, The Aga Khan University, Karachi. All the data will be saved on a computer with a password and access was allowed to only primary and corresponding authors.

Collected data were summarized as mean ± standard deviation (SD), median [IQR] or frequency (%). Patients were stratified based on gender and age and CSF investigations and imaging results were compared with the help of Chi-square test/Fisher's exact test at 0.05 level of significance. The analysis was done using IBM SPSS version 21.

## Results

### Demographics

A total of 39 patients were included in the study, with 33 males (84.6%) and 6 females (15.4%) The median age of the patients was 34 years. The median duration of symptoms was three days while the median length of hospital stay was two days (Table 1). Most cases were reported during May to September with the highest number seen in July (Fig 1).

### Clinical presentation

The most frequent presenting complaints included fever found in 37 patients (94.9%) followed by headache in 28 patients (71.8%), nausea and vomiting in 27 patients (69.2%), and seizures in 10 patients (25.6%) (Table 1).

Table 1. Distribution of demographic characteristics and clinical presentation of patients admitted with primary amoebic meningo-encephalitis.

| Characteristics | Summary Statistics |
|---|---|
| **Total (N)** | **39** |
| **Gender %** | |
| Male | 33 (84.6) |
| Female | 6 (15.4) |
| **Median age (years)** | 34 |
| 18 to 35 years | 22 (56.4) |
| >35 years | 17 (43.6) |
| **Median length of stay (days)** | 2 (2–3) |
| **Median duration of symptoms (days)** | 3 (2–3) |
| **Presenting symptoms and signs** | |
| Fever | 37 (94.9) |
| Nausea/vomiting | 27 (69.2) |
| Drowsiness | 3 (7.7) |
| Headache | 28 (71.8) |
| Seizure | 10 (25.6) |
| Decreased consciousness | 25 (64.1) |
| Irritability | 18 (46.1) |
| **Median Glasgow Coma Scale (GCS) on arrival** | 12 (10–13) |

## Treatment

The treatment for PAM was initiated in 39 patients which included intravenous amphotericin B 1.5 mg/kg every 12 hours and 1.5 mg intrathecal, intravenous azithromycin 10 mg/kg once daily (OD), Intravenous fluconazole 800mg once daily (OD), rifampin per oral 600mg once daily (OD), chlorpromazine 500mg Q6H (up to 15 to 20 mg per kg), and per oral miltefosine 50mg every eight hourly per day in 27 patients. The remaining 10 patients did not receive miltefosine but received the rest of the treatment regimen. All patients received IV Dexamethasone 0.6mg per kg in divided doses along with the rest of the treatment. Out of these 39 patients only one patient survived.

The patient who survived was a 24 years old gentleman with had no comorbid, presented with fever and vomiting for two days, along with altered sensorium for one day and generalized tonic colonic seizure on arrival in emergency. There was no history of water recreation.

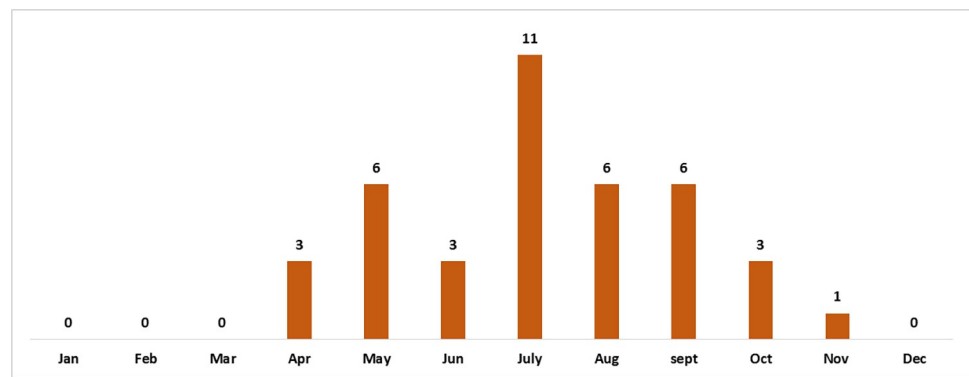

Fig 1. Distribution of admitted cases of primary amoebic meningo-encephalitis by calendar months over 2010 to 2021.

On arrival he was hemodynamically stable, GCS was 9/15, neck was rigid, with brisk reflexes and bilateral extensor plantars.

Initial MRI on 6/9/14 showed no acute infarct, intracranial hematoma or abnormal parenchymal enhancement. There was Loss of CSF FLAIR signals with subtle meningeal enhancement, suggestive of leptomeningeal inflammation. The initial CSF DR done on first day showed low Glucose (29 mg/dl), high protein (213 mg/dl), high RBCs 642/cu mm, and raised WBC count (523/cu mm) with 80% Lymphocytes. CSF PCR and culture was positive for *Naegleria Fowleri*. Patient was started on 1). Miltefosine per oral 50mg Q8H, 2) Intravenous. Amphotericin B 1.5 mg/kg every 12 hours and 1.5 mg intrathecal per day, 3) per oral Azithromycin 750mg once daily (OD), per oral Fluconazole 800mg on day one then followed by 400mg once daily (OD), Rifampin per oral 600mg daily, IV Dexamethasone 0.6mg per kg per day. Patient was also empirically started on meningitic doses of Intravenous Meropenem 200mg Q8H and Intravenous Vancomycin 750mg Q6H. Intensive care unit team was taken on board due to low GCS and respiratory distress but did not require invasive ventilation. A repeat CSF DR done on third day of treatment showed a Glucose of 29 mg/dl, Protein 119 mg/dl, Tlcs 856 /cu mm with 60% Lymphocytes.

The hospital course of the patient got complicated with acute liver injury and kidney injury leading to the complete renal shutdown. Patient was started on hemodialysis which continued from 27/9/2014 till 3/10/2014 while all the antibiotics were also continued. On 28/9/2014 GCS started to improve from 8/15 to 13/15 on 4/10/2014.

Last MRI was done on 21/9/2014 which showed (Abnormal signal intensity area in left frontal lobe not showing diffusion restriction on DWI however showing focal hemorrhage. There was also diffuse abnormal signals in bilateral caudate nuclei showing diffusion restriction on DWI, appearing low on ADC and showing foci of hemorrhage. Findings were most likely due to vasculitic infarction with hemorrhagic conversion.

PAM protocol was stopped on 4/10/2014 after total of 28 days of treatment through combined decision of Infectious Disease team and Neurology team based on improved patient's condition and risk of additional drug Side effects.

Patient was discharged home on 7/10/2014 with a 31 days long hospital course with a GCS of 15/15 but with mild right sided residual weakness. On follow up clinic visit after one and half month, the patient had marked improvement with independent function of daily living.

## Investigations and outcomes

Of the 39 patients that underwent lumbar puncture, 27 patients (69.2%) had positive motile trophozoites on CSF wet preparation microscopy, 18 patients (46.2%) had a positive culture, and 10 patients (25.6%) had a positive PCR. Cerebrospinal fluid (CSF) analysis resembled bacterial meningitis with elevated white blood cell counts with predominantly neutrophils (median, 3000 cells/μL [range, 1350–7500 cells/μL]), low glucose levels median, 14 mg/dL [range, 1–92 mg/dL]), and elevated protein levels (median, 344 mg/dL [range, 289–405 mg/dL]) (**Table 2**). Imaging results were abnormal in approximately three-fourths of the patients.

Cerebral edema (66.6%), hydrocephalus (25.6%) and cerebral infarction (12.8%) were the most common findings. Only one patient managed to survive (**Table 2**).

## Role of gender

There were no statistically significant differences between male and female patients in terms of CSF investigations and imaging results (**Table 3**). However, the disease has a predilection towards male gender. There was a tendency of higher positive wet mount among male patients with frequency of 72.7% vs. 50.0%; p = 0.348 for male and female patients, respectively.

**Table 2. Distribution of CSF and lab investigations, and imaging results of patients admitted with primary amoebic meningo-encephalitis.**

| Characteristics | Summary Statistics (%) |
|---|---|
| **Total (N)** | **39** |
| **CSF Investigations** | |
| Positive Wet Mount | 27 (69.2) |
| Positive Culture | 18 (46.2) |
| Positive PCR | 10 (25.6) |
| Median WBC (units) | 3000 (1350–7500) |
| Median LYMP (units) | 20 (10–25) |
| Median POLY (units) | 80 (75–90) |
| Median Protein (units) | 344 (289–405) |
| Median Glucose (units) | 14 (5–45) |
| **Lab Investigations** | |
| Median HB (units) | 13.7 (11.7–15.2) |
| Median PLT (units) | 207 (178–236) |
| Median CREAT (units) | 1.1 (0.9–1.4) |
| Median BICARB (units) | 17.1 (15.5–18.3) |
| Median SODIUM (units) | 136 (132–140) |
| Median SGPT (units) | 27 (21–44) |
| **Imaging** | |
| **1st CT Performed** | **34 (87.2)** |
| Basal Enhancement | 3 (8.8) |
| Infarction | 3 (8.8) |
| Cerebritis | 1 (2.9) |
| Lepto Enhance | 5 (14.7) |
| Uncus Herniation | 0 (0) |
| Hydroceph | 6 (17.6) |
| Cereb Edema | 13 (38.2) |
| Ventriculitis | 1 (2.9) |
| Midline Shift | 0 (0) |
| Tonsil Herniation | 2 (5.91) |
| **2nd CT Performed** | **17 (43.6)** |
| Basal Enhancement | 0 (0) |
| Infarction | 1 (5.9) |
| Cerebritis | 0 (0) |
| Lepto Enhance | 0 (0) |
| Uncus Herniation | 0 (0) |
| Hydroceph | 3 (17.6) |
| Cereb Edema | 15 (88.2) |
| Ventriculitis | 0 (0) |
| Midline Shift | 0 (0) |
| Tonsil Herniation | **6 (35.3)** |
| **MRI performed** | 9 (23.0) |
| Basal Enhancement | 7 (77.8) |
| Infarction | 1 (11.1) |
| Cerebritis | 0 (0) |
| Uncus Herniation | 0 (0) |
| Hydroceph | 1 (11.1) |

(*Continued*)

**Table 2.** (Continued)

| Characteristics | Summary Statistics (%) |
|---|---|
| Cereb Edema | 2 (22.2) |
| Ventriculitis | 0 (0) |
| Midline Shift | 0 (0) |
| Tonsil Herniation | 2 (22.2) |
| **Outcome** | |
| Died | 38 (97.4) |
| Alive | 1 (2.6) |
| **Collective Imaging results** | |
| Basal Enhancement | 9 (23.0) |
| Infarction | 5 (12.8) |
| Cerebritis | 1 (2.6) |
| Lepto Enhance | 5 (12.8) |
| Uncus Herniation | 0 (0) |
| Hydrocephalus | 10 (25.6) |
| Cerebral Edema | 26 (66.6) |
| Ventriculitis | 1 (2.6) |
| Midline Shift | 0 (0) |
| Tonsil Herniation | 9 (23.1) |

While imaging results showed higher tendency of cerebral edema among female patients with frequency of 60.6% vs. 100.0%; p = 0.081 for female and male patients, respectively (**Table 3**).

## Role of age

The CSF investigations showed significantly higher proportion of positive CSF cultures among patients aged $> 35$ years with frequency of 70.6% vs. 27.3%; p = 0.007 for patients with $>35$

**Table 3. Comparison of CSF investigations and imaging results of patients admitted with primary amoebic meningo-encephalitis by gender.**

| | Male | Female | P-value |
|---|---|---|---|
| **CSF Investigations** | | | |
| Positive Wet Mount | 24 (72.7) | 3 (50) | 0.348 |
| Positive Culture | 17 (51.5) | 1 (16.7) | 0.19 |
| Positive PCR | 8 (24.2) | 2 (33.3) | 0.636 |
| **Combine Imaging results** | | | |
| Basal Enhancement | 7 (21.2) | 2 (33.3) | 0.607 |
| Infarction | 4 (12.1) | 1 (16.7) | >0.999 |
| Cerebritis | 1 (3) | 0 (0) | >0.999 |
| Lepto Enhance | 5 (15.2) | 0 (0) | 0.574 |
| Uncus Herniation | 0 (0) | 0 (0) | - |
| Hydroceph | 8 (24.2) | 2 (33.3) | 0.636 |
| Cereb Edema | 20 (60.6) | 6 (100) | 0.081 |
| Ventriculitis | 1 (3) | 0 (0) | >0.999 |
| Midline Shift | 0 (0) | 0 (0) | - |
| Tonsil Herniation | 7 (21.2) | 2 (33.3) | 0.607 |

**Table 4. Comparison of CSF investigations and imaging results of patients admitted with primary amoebic meningo-encephalitis by age.**

|  | 18 to 35 years | >35 years | P-value |
|---|---|---|---|
| **CSF Investigations** | | | |
| Positive Wet Mount | 14 (63.6) | 13 (76.5) | 0.389 |
| Positive Culture | 6 (27.3) | 12 (70.6) | 0.007 |
| Positive PCR | 6 (27.3) | 4 (23.5) | >0.999 |
| **Combine Imaging results** | | | |
| Basal Enhancement | 6 (27.3) | 3 (17.6) | 0.704 |
| Infarction | 4 (18.2) | 1 (5.9) | 0.636 |
| Cerebritis | 0 (0) | 1 (5.9) | 0.436 |
| Lepto Enhance | 1 (4.5) | 4 (23.5) | 0.147 |
| Uncus Herniation | 0 (0) | 0 (0) | - |
| Hydroceph | 5 (22.7) | 5 (29.4) | 0.721 |
| Cereb Edema | 16 (72.7) | 10 (58.8) | 0.497 |
| Ventriculitis | 0 (0) | 1 (5.9) | 0.436 |
| Midline Shift | 0 (0) | 0 (0) | - |
| Tonsil Herniation | 8 (36.4) | 1 (5.9) | 0.052 |

years and 18 to 35 years of age, respectively. Similarly, imaging investigations showed higher proportion of tonsillar herniation among patients aged 18 to 35 years with frequency of 36.4% vs. 5.9%; p = 0.052 for patients with 18 to 35 years and >35 years of age, respectively (**Table 4**).

## Discussion

The progression of PAM disease is very rapid, with extremely high mortality rates. In this study our aim was to be identifying the spectrum of symptoms with which patients present to the hospital, reviewing the clinical course, duration of symptoms and lab parameters, specifically looking at the factors that predict poor outcome and the factors that delay the disease process and ultimately mortality. Our data showed that almost all features of *Naegleria Fowleri* induced PAM are like the cases found in other parts of the world. Common presenting features include fever, headache, vomiting and at late presentation, seizures with or without altered neurological status [9].

In our study, peak incidence was reported during the months of May to September which coincides with a hotter temperature [10]. Similar to earlier reports, only a small proportion of patients had confirmed history of swimming or water recreational activity which is strongly linked with the pathogenesis of *Naegleria* PAM [11], while the other portion of patients had no such history. Virtually all patients were Muslims, as a result the infection probably transpired through ablution in conjunction with tap water.

The highest number of PAM cases in the USA were reported in children younger than 14 years, but in Pakistan, most cases are reported in adults aged 26–45 years, which may possibly be due to a genetically unique strain in Pakistan. More research is needed to look at the genome sequence for a likely emerging resistant strain in Pakistan [12].

The male predominance seen in our sample and overall reported cases may be related to activities of water exposure from different sources compared to females with limited exposure or may be predisposed due to sex-linked hormones as has been hypothesized for other infections such as Entamoeba histolytica liver abscesses [13].

Poor healthcare system, and unavailability of potent drugs to manage this disease is also a huge health risk for our country [14, 15]. Unfortunately, our data also revealed that *Naegleria*

*Meningitis* has high mortality rate (97.4%) which was attributed to late presentation, delay in accurate diagnosis, and short incubation period leading to death in 72 hours of first symptoms appearance [16–18].

It is a single center audit for the duration of more than 10 years of 39 *Naegleria* cases. Although almost all the findings are consistent with earlier similar work, the source of spread is somewhat different in our part of the world as compared to west.

Besides working on traditional mode of spread of *Naegleria* which includes swimming/recreational activities in contaminated water, water reservoirs for domestic/daily usage in underground tanks should also be considered a potential source. Like other seasonal pandemics such as dengue, COVID, malaria, flu, hemorrhagic fever, we should also prepare ourselves for *Naegleria* surge and use social, electronic, and print media to educate our community to efficiently combat these fatal diseases. Efforts should be made at government level to improve living infrastructure in provision of clean and safe chlorinated water to all and to improve healthcare facilities. Chlorination is the most extensively used disinfectant for water treatments due to its low cost, ease to produce, store, transport and use as well as its high oxidizing potential. Usually used as chlorine gas, sodium or calcium hypochlorite, it provides a minimum level of residual disinfectant able to prevent microbial recontamination [19]. Chlorine dioxide have been as well reported by Dupuy et al., (2014) for its efficacy to inhibit three different FLA strains [20]. The Karachi Water and Sewerage Board (KWSB) is the principal agency in Karachi, Pakistan which oversees issues with water distribution and chlorination in the city, has been ordered by the health authorities to ensure that the municipal water supply is adequately chlorinated with the WHO recommendations, and preventive actions have also been developed to stop the spread of this sickness [21].

## Limitations

Small sample size as well as single-center study.

## Conclusions

PAM is a fatal illness with limited treatment success. Treating Physicians should have a high suspicion for PAM in the setting of acute meningo-encephalitis with rapidly progressive symptoms with negative CSF results for bacteria and common viruses. Early diagnosis and prompt initiation of treatment may improve the survival of the patients and reduce mortality.

## Author Contributions

**Conceptualization:** Shakeel Ur Rehman, Salman Farooq, Muhammad Bilal Tariq, Mohammad Wasay.

**Data curation:** Shakeel Ur Rehman, Nosheen Nasir, Sobia Masood, Musa Karim.

**Formal analysis:** Salman Farooq, Muhammad Bilal Tariq, Nosheen Nasir, Sobia Masood, Musa Karim.

**Investigation:** Shakeel Ur Rehman, Nosheen Nasir.

**Methodology:** Shakeel Ur Rehman, Nosheen Nasir, Sobia Masood, Musa Karim.

**Project administration:** Nosheen Nasir.

**Software:** Musa Karim.

**Supervision:** Shakeel Ur Rehman, Salman Farooq, Muhammad Bilal Tariq, Nosheen Nasir, Mohammad Wasay.

**Visualization:** Mohammad Wasay.

**Writing – original draft:** Salman Farooq.

**Writing – review & editing:** Shakeel Ur Rehman, Salman Farooq, Nosheen Nasir.

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
