## [Decision Letter · Decision Letter 0]

3 May 2023

PONE-D-23-06572Clinical Manifestations and Outcome of Patients with Primary Amoebic Meningoencephalitis in Pakistan. A Single Center ExperiencePLOS ONE

Dear Dr. Farooq,

Thank you for submitting your manuscript to PLOS ONE. After careful consideration, we feel that it has merit but does not fully meet PLOS ONE’s publication criteria as it currently stands. Therefore, we invite you to submit a revised version of the manuscript that addresses the points raised during the review process.

The reviewers have made comments that will improve this manuscript. I agree that adding information on the patient that survived would be wonderful since there are so few reports of PAM survivors.

We look forward to receiving your revised manuscript.

Kind regards,

Kelli L. Barr, Ph.D.

Academic Editor

PLOS ONE

Journal Requirements:

 Whilst you may use any professional scientific editing service of your choice, PLOS has partnered with both American Journal Experts (AJE) and Editage to provide discounted services to PLOS authors. Both organizations have experience helping authors meet PLOS guidelines and can provide language editing, translation, manuscript formatting, and figure formatting to ensure your manuscript meets our submission guidelines. To take advantage of our partnership with AJE, visit the AJE website (http://aje.com/go/plos) for a 15% discount off AJE services. To take advantage of our partnership with Editage, visit the Editage website (www.editage.com) and enter referral code PLOSEDIT for a 15% discount off Editage services. If the PLOS editorial team finds any language issues in text that either AJE or Editage has edited, the service provider will re-edit the text for free.

 A clean copy of the edited manuscript (uploaded as the new *manuscript* file).

Reviewers' comments:

Reviewer's Responses to Questions

**Comments to the Author**

1. Is the manuscript technically sound, and do the data support the conclusions?

Reviewer #1: Yes

Reviewer #2: Yes

Reviewer #3: Partly

2. Has the statistical analysis been performed appropriately and rigorously? 

Reviewer #1: I Don't Know

Reviewer #2: Yes

Reviewer #3: Yes

3. Have the authors made all data underlying the findings in their manuscript fully available?

Reviewer #1: Yes

Reviewer #2: Yes

Reviewer #3: Yes

4. Is the manuscript presented in an intelligible fashion and written in standard English?

Reviewer #1: Yes

Reviewer #2: Yes

Reviewer #3: Yes

5. Review Comments to the Author

Reviewer #1: 1. The scientific name of Naegleria fowleri should be correctly addressed as italics through the manuscript.

2. IRB approval number should be clearly provided.

3. Table 1: GCS means Glasgow Coma Scale? If yes, please describe it clearly.

4. In Treatment: Why only 37 patients, not 39? Detailed description for one survivor would be interesting.

5. Were there any specific reasons to classify the patients into two groups, 18-35 and over 35 years old, by age?

6. Tables 3 and 4: The numbers of patients based on the classification could be included.

7. Table 4: Few values such as infarction, lepto enhance, and tonsil herniation in combined imaging results showed differences between the two groups, even though the number of patients were not high. 8. Discussion on the differences in the discussion section would be appreciated.

9. Line 155: Proper references should be added.

10. Considering the main points of this manuscript were the clinical features in PAM patients, discussion could be improved further by comparing the previously reported clinical features in Pakistan and other global areas.

11. English could be improved. English editing is recommended.

Reviewer #2: According to my expertise, this article fulfills all the requirements mentioned in the PLOSOne guidelines.

The researchers of the article have done a rigorous analysis on their sample population and provided with the results in written and table forms as well which makes it easier to comprehend.

The method and material portion upheld the ethical code and was approved by IRB.

However, there are few changes that I would like to suggest.

1.Out of these 39 patients only one patient who survived received all

105 these treatment for total of 24 days and was stopped by infectious disease team.

Suggestion a: Please mention the clinical criteria used by the infectious disease team to decide when to stop treatment for this patient. This information can provide further insight into prognostic factors and recovery criteria that can be used by doctors managing this disease in future.

Suggestion b: Kindly also mention any demographic or clinical presentation factors that differ between this patient and the other deceased patients, as the mortality rate of this disease is high. Elaborating on why this particular patient may have survived could help the medical community identify potential factors that contribute to better outcomes. This information could inform further studies that could hypothesize about the causal relationship between these factors and disease severity.

2.Efforts should be made at government level to improve living infrastructure in provision

of clean and safe chlorinated water to all and to improve healthcare facilities.

Suggestion: kindly provide a reference for the claim that chlorinated water can reduce the spread of Naegleria, and if so, how? This information could provide guidelines for the prevention of this infection. Alternatively, could you mention any already-published guidelines used by developed countries to prevent the spread of this disease?"

3.

Reviewer #3: First of all, I want to thank to all of the authors of the manuscript. This study needs to be revised with more detailed and proper written language.

Naegleria fowleri, the causative agent of primary amoebic meningoencephalitis (PAM), is a free-living amoeba. It is a water-borne infection usually detected in children and young people with healthy immune systems who swim, dive and perform activities in fresh and hot springs. It is an important study in terms of addressing the Clinical Symptoms and Outcomes of Patients with Primary Amoebic Meningoencephalitis in Pakistan. The points mentioned below need to be considered in more detail.

- "Naegleria fowleri" or "Naegleria" needs to be corrected in italics throughout the manuscript.

- -In line 56, it is stated that the transmission route of N. fowleri can not only be through contaminated water, such as swimming in freshwater lakes, rivers, canals, and swimming pools, but also "dry infection" with dust in the air. It would be better to mention this transmission route here.

- In line 93, it should also mention the decrease in consciousness about the clinical symptoms that are the backbone of the study.

- The term "GCS" is not mentioned anywhere in the manuscript.

- In line 102, the spelling of "2 gm/day" should be corrected.

- In line 104, explaining the drug treatment complex administered for a surviving patient in more detail would be more beneficial.

- It is interesting t that 27 patients were positive by microscopy, while only ten were positive by PCR. Besides, the PCR method is more sensitive and specific, and PCR results are essential in confirming the microscopy. It would be better if you could explain how you evaluate this situation.

- Cerebral edema is 66.7% in line 114 and 66.6% in the table; whichever is correct should be corrected.

- Looking at the table, tonsil herniation (23.1%) is among the most common ones.

- "While imaging results showed higher tendency of cerebral edema among female patients with frequency of 60.6% vs. 100.0%; p=0.081 for female and male patients, respectively" in line 122 This sentence should be reviewed. Because while 100% cerebral edema is seen in women in table 3, it is expressed as 60.6% in men.

- In the discussion section, the clinical symptoms of the patients and the treatment options given should be discussed in more detail. For example, it should be discussed compared to clinical symptoms and treatments in a single case (https://doi.org/10.1007/s11686-021-00514-0).

6. PLOS authors have the option to publish the peer review history of their article (what does this mean?). If published, this will include your full peer review and any attached files.

Reviewer #1: No

Reviewer #2: **Yes: **Syeda Maria Hassan

Reviewer #3: **Yes: **Mehmet AYKUR

---

## [Author Response · Author response to Decision Letter 0]

19 Jul 2023

Reviewer #1: 1. The scientific name of Naegleria fowleri should be correctly addressed as italics through the manuscript. Thank you for this feedback. We have corrected the name Naegleria fowleri in italics through the manuscript. 2. IRB approval number should be clearly provided. Thank you for this comment. The IRB file is attached to the submission. For clarification, we have also added the IRB # ( 2021-6074-18800) to the manuscript.For Information: Tel: +92 21 3493 0051 Ext: 4988|2445 Email: erc.pakistan@aku.edu 3. Table 1: GCS means Glasgow Coma Scale? If yes, please describe it clearly. Thank you for your comment. We have added Glasgow Coma Scale to describe GCS clearly in the table. 4. In Treatment: Why only 37 patients, not 39? Detailed description for one survivor would be interesting. We thank the reviewer for this comment. We have corrected the number of patients to 39. We have also added additional detail on the patient that survived. 5. Were there any specific reasons to classify the patients into two groups, 18-35 and over 35 years old, by age? The patient groups were divided into two age groups to see if any difference in incidence, diagnostic,radio-logical and clinical features in younger patients compared to older patients. The age cut off was arbitrarily decided. 6. Tables 3 and 4: The numbers of patients based on the classification could be included.Thank you for this comment. We have added the total number of patients with the qualifications. 7. Table 4: Few values such as infarction, lepto enhance, and tonsil herniation in combined imaging results showed differences between the two groups, even though the number of patients were not high.Thank you for your comment. The differences were not statistically significant at the prespecified level of significance set at 0.05.8. Discussion on the differences in the discussion section would be appreciated. 9. Line 155: Proper references should be added. 10. Considering the main points of this manuscript were the clinical features in PAM patients, discussion could be improved further by comparing the previously reported clinical features in Pakistan and other global areas. 11. English could be improved. English editing is recommended. Thank you for the comment. We have rectified. Reviewer #2: According to my expertise, this article fulfills all the requirements mentioned in the PLOSOne guidelines.The researchers of the article have done a rigorous analysis on their sample population and provided with the results in written and table forms as well which makes it easier to comprehend. The method and material portion upheld the ethical code and was approved by IRB. However, there are few changes that I would like to suggest. 1.Out of these 39 patients only one patient who survived received all 105 these treatment for total of 24 days and was stopped by infectious disease team. Suggestion a: Please mention the clinical criteria used by the infectious disease team to decide when to stop treatment for this patient.This information can provide further insight into prognostic factors and recovery criteria that can be used by doctors managing this disease in future. Suggestion b: Kindly also mention any demographic or clinical presentation factors that differ between this patient and the other deceased patients, as the mortality rate of this disease is high. Elaborating on why this particular patient may have survived could help the medical community identify potential factors that contribute to better outcomes. This information could inform further studies that could hypothesize about the causal relationship between these factors and disease severity. Thank you for your comment. We have added the clinical course of the patient to better convey their treatment. It is difficult to analyze any major differences between this patient and the patients that did not survive. We have added some of our thoughts. 2.Efforts should be made at government level to improve living infrastructure in provision of clean and safe chlorinated water to all and to improve healthcare facilities. Suggestion: kindly provide a reference for the claim that chlorinated water can reduce the spread of Naegleria, and if so, how? This information could provide guidelines for the prevention of this infection. Alternatively, could you mention any already-published guidelines used by developed countries to prevent the spread of this disease? Thank you for this comment. We have added references and a paragraph in the discussion to support this statement. Reviewer #3: First of all, I want to thank to all of the authors of the manuscript. This study needs to be revised with more detailed and proper written language. Naegleria fowleri, the causative agent of primary amoebic meningoencephalitis (PAM), is a freeliving amoeba. It is a water-borne infection usually detected in children and young people with healthy immune systems who swim, dive and perform activities in fresh and hot springs. It is an important study in terms of addressing the Clinical Symptoms and Outcomes of Patients with Primary Amoebic Meningoencephalitis in Pakistan. The points mentioned below need to be considered in more detail. - "Naegleria fowleri" or "Naegleria" needs to be corrected in italics throughout the manuscript. Thank you for your suggestion. We have corrected it in the manuscript. - -In line 56, it is stated that the transmission route of N. fowleri can not only be through contaminated water, such as swimming in freshwater lakes, rivers, canals, and swimming pools, but also "dry infection" with dust in the air. It would be better to mention this transmission route here.Thank you for this comment. We have added this as a statement. - In line 93, it should also mention the decrease in consciousness about the clinical symptoms that are the backbone of the study. Thank you for the comment. We have rectified. - The term "GCS" is not mentioned anywhere in the manuscript.The full form of Glasgow Coma Scale has been added on first mention, and abbreviation as well. Thank you for the comment. We have added the full form. - In line 102, the spelling of "2 gm/day" should be corrected.Thank you for pointing this out. We have corrected this. - In line 104, explaining the drug treatment complex administered for a surviving patient in more detail would be more beneficial.Thank you for your comment. We have added the clinical course of the patient to better convey their treatment. - It is interesting t that 27 patients were positive by microscopy, while only ten were positive by PCR. Besides, the PCR method is more sensitive and specific, and PCR results are essential in confirming the microscopy. It would be better if you could explain how you evaluate this situation. PCR method was introduced at our hospital at 2014 and before this it was diagnosed on wet mount. Also when it was diagnosed on Wet mount PCR was not done in that case. In highly suspected case like short history and early deterioration PCR was done. This has been clarified in the manuscript. - Cerebral edema is 66.7% in line 114 and 66.6% in the table; whichever is correct should be corrected.The correct value is 66.6 and has been corrected.- Looking at the table, tonsil herniation (23.1%) is among the most common ones.- "While imaging results showed higher tendency of cerebral edema among female patients with frequency of 60.6% vs. 100.0%; p=0.081 for female and male patients, respectively" in line 122 This sentence should be reviewed. Because while 100% cerebral edema is seen in women in table 3, it is expressed as 60.6% in men. This was miswritten and has been corrected. - In the discussion section, the clinical symptoms of the patients and the treatment options given should be discussed in more detail. For example, it should be discussed compared to clinical symptoms and treatments in a single case (https://doi.org/10.1007/s11686-021-00514-0). Thank you for the comment. We have rectified.

---

## [Decision Letter · Decision Letter 1]

8 Aug 2023

Clinical Manifestations and Outcome of Patients with Primary Amoebic Meningoencephalitis in Pakistan. A Single Center Experience

PONE-D-23-06572R1

Dear Dr. Farooq,

We’re pleased to inform you that your manuscript has been judged scientifically suitable for publication and will be formally accepted for publication once it meets all outstanding technical requirements.

Kind regards,

Kelli L. Barr, Ph.D.

Academic Editor

PLOS ONE

Additional Editor Comments (optional):

Reviewers' comments:

Reviewer's Responses to Questions

**Comments to the Author**

1. If the authors have adequately addressed your comments raised in a previous round of review and you feel that this manuscript is now acceptable for publication, you may indicate that here to bypass the “Comments to the Author” section, enter your conflict of interest statement in the “Confidential to Editor” section, and submit your "Accept" recommendation.

Reviewer #2: All comments have been addressed

Reviewer #3: (No Response)

2. Is the manuscript technically sound, and do the data support the conclusions?

Reviewer #2: Yes

Reviewer #3: (No Response)

3. Has the statistical analysis been performed appropriately and rigorously? 

Reviewer #2: Yes

Reviewer #3: (No Response)

4. Have the authors made all data underlying the findings in their manuscript fully available?

Reviewer #2: Yes

Reviewer #3: (No Response)

5. Is the manuscript presented in an intelligible fashion and written in standard English?

Reviewer #2: Yes

Reviewer #3: (No Response)

6. Review Comments to the Author

Reviewer #2: (No Response)

Reviewer #3: (No Response)

7. PLOS authors have the option to publish the peer review history of their article (what does this mean?). If published, this will include your full peer review and any attached files.

Reviewer #2: **Yes: **Syeda Maria Hassan

Reviewer #3: **Yes: **Mehmet Aykur

---

## [Editor Report · Acceptance letter]

11 Aug 2023

PONE-D-23-06572R1 

Clinical Manifestations and Outcome of Patients with Primary Amoebic Meningoencephalitis in Pakistan. A Single-Center Experience 

Dear Dr. Farooq:

I'm pleased to inform you that your manuscript has been deemed suitable for publication in PLOS ONE. Congratulations! Your manuscript is now with our production department. 

Kind regards, 

on behalf of

Dr. Kelli L. Barr 

Academic Editor

PLOS ONE